# Modulation of cellular adhesion, contractility, and migration by MiuA: A comprehensive analysis of its biomechanical impact

Carsten Alexander Baltes[1], Friederike Nolle[1,2], Kathi Michèle Kaiser[1], Erbara Gjana[1], Kristin Sander[1], Karin Jacobs[1,3], Rhoda Joy Hawkins[4,5], Franziska Lautenschläger[1,3]*

**1** Experimental Physics, Saarland University, Saarbrücken, Germany, **2** Department of Electrical Engineering, Trier University of Applied Science, Schneidershof, Trier, Germany, **3** Centre of Biophysics, Saarland University, Saarbrücken, Germany, **4** School of Mathematical and Physical Sciences, University of Sheffield, Sheffield, United Kingdom, **5** African Institute for Mathematical Sciences, Accra, Ghana

* f.lautenschlaeger@physik.uni-saarland.de

## Abstract

Cellular adhesion and contractility are essential for cell movement. In this study, we investigated the effects of actin stabilization on adhesion properties, contractility, and cell migration. For this, we used the recently synthesized actin stabilizer miuraenamide A (MiuA), which has been discussed as a more reliable alternative to the otherwise commonly used actin stabilizer jasplakinolide. We investigated the number and size of focal adhesions in RPE-1 cells and used single-cell force spectroscopy to evaluate the adhesion properties of those cells after MiuA treatment. We showed that MiuA increases the number of focal adhesions while decreasing their size and reduces adhesion energy and force. Additionally, we investigated its effects on the contractility of RPE-1 cells by measuring their contractile energy using pattern-based contractility screening (PaCS). We found no significant change in contractility after MiuA treatment. Finally, we confined RPE-1 cells in PDMS microchannels and analyzed their migration after treatment with MiuA, showing that neither their speed nor their persistence is affected by MiuA. To check that these effects are not specific to RPE-1 cells, we also analyzed the effects of MiuA treatment in MEF cells and neutrophils. Both MEF cells and neutrophils showed the same results as the RPE-1 cells. Our measurements indicate that, although altering focal adhesions significantly reduces adhesion, it does not impact cell contractility. This finding also clarifies why amoeboid migration, which operates independently of adhesion, remains unaffected. Additionally, it explains the previously observed reduction in mesenchymal migration, which relies on adhesion-based mechanisms.

**Data availability statement:** All data, p-values and python codes are availabe from the database "figshare" via the following link: https://doi.org/10.6084/m9.figshare.26862634.v3 Full citation: Lautenschläger, Franziska (2024). Data, p-values and python3 codes for manuscript "Actin length in cellular adhesion and contractility". figshare. Dataset. https://doi.org/10.6084/m9.figshare.26862634.v3

**Funding:** This research was supported by the German Research Foundation (SFB1027) and large-instrument funding under grant number INST 256/542-1 FUGG. The founders had no role in study design, data collection, and analysis, decision to publish, or preparation of the manuscript. There was no additional funding received for this study.

**Competing interests:** The authores have declared that no competing interests exist.

## Introduction

The ability of cells to migrate is essential for a multitude of cellular processes. In general cells show two distinct modes of migration. They can either migrate in an adhesion-based manner, so called mesenchymal migration, or rely on friction forces and contractility to propel themselves forward, so called amoeboid migration. These two modes are not exclusive as some cells can switch between them and even display modes of migration that have characteristics from both [1]. For our research we decided to work with three different cell lines: RPE-1 (Retinal Pigment Epithelial) cells, MEF (Mouse Embryonic Fibroblasts) cells, and neutrophils. RPE-1 cells are a commonly used cell line that is capable of adhesion and migration. However, we specifically chose RPE-1 cells for their ability to switch between mesenchymal and amoeboid migration [1]. In a similar way, MEF cells are an adherent cell line and perform predominantly mesenchymal migration, therefore allowing us to compare them to RPE-1 cells. Lastly, neutrophils are non-adherent cells, which are differentiated from HL60 cells. They are used as a model for amoeboid migration [2] and serve our research as a control system to validate our findings in amoeboid migration of RPE-1 cells.

In their study of cell migration, Lämmermann et al. present three main processes: Actin polymerization, cellular adhesion, and cell contractility [3].

In all three processes, the cytoskeletal protein actin plays a crucial role. It is therefore a prominent target when investigating the migration behavior of cells. A variety of compounds are commonly used to modify the structure and dynamics of the actin network in cells. While these compounds have enabled the study of the migration behavior of cells in general, their effects on the underlying migration processes mentioned above remain unclear.

Substances that depolymerize the actin network, such as cytochalasin D or latrunculin A, have been used to reliably study the effects of a disassembled actin network on cellular functions, including migration [4–10]. Nevertheless, the stabilization of actin filaments has proven to be a significant challenge. The two most prominent actin stabilizing agents are phalloidin and jasplakinolide. The first is unable to pass through the cell membrane [11], while the effectiveness of the second is dependent on the concentration used and the duration of exposure, with optimal results requiring precise calibration [12].

Nevertheless, the actin stabilization compound jasplakinolide has been used in various studies. It has been shown to reduce migration in MDA-MB-231 and MCF7 cells in a dose dependent manner [10,13] but shows the contrary effect on A549 cells [10]. Additionally, experiments on cellular growth and apoptosis showed that jasplakinolide not only reduces cellular growth but is also capable of inducing apoptosis [14,15], as well as actin depolymerization [16].

The versatile use of jasplakinolide reflects the interest in the community of the interplay between actin stabilization and cellular mechanics. However, jasplakinolide remained the only available actin stabilizer suitable for live cell experiments for a long time.

Recently, a new compound, miuraenamide A (MiuA), has been used in studies to stabilize actin filaments in living cells and in vitro [17–21]. This new tool allows investigation of the effects of actin stabilization in migration and its underlying mechanisms.

MiuA achieves this stabilizing effect by shifting the conformation of F-actin monomers [17]. MD simulations performed by Wang et al. showed that this change in conformation blocks the binding site of cofilin, an actin depolymerizing factor, which ultimately results in a decreased depolymerization rate of actin filaments and leads to an increased length of actin filaments. This effect has also been verified in vitro by Wang et al. [17]. Additionally, Gegenfurtner et al. tested the transcriptional effects of MiuA [22]. They report a net positive effect on the following categories of genes: Cellular adhesion and spreading, smooth muscle contraction and regulation of lamellipodium assembly.

In an earlier study, we demonstrated that MiuA reduces actin filament dynamics, thereby altering the first of the migration processes discussed by Lämmermann et al.: the polymerization rate. Additionally, we observed that MiuA treatment reduced mesenchymal migration in RPE-1 and MEF cells in a 2D unconfined environment [23]. Following our previous work, we wanted to understand the effects of the actin stabilizing compound MiuA on the remaining two migration mechanisms: adhesion and contractility, on amoeboid as well as mesenchymal cell migration. We therefore performed adhesion and contractility measurements on RPE-1 and on MEF cells, and employed amoeboid migration assays in PDMS microchannels using RPE-1 cells and neutrophils.

In this new study we showed that MiuA stabilizes and elongates actin filaments in RPE-1 and MEF cells and also affects the number and size of focal adhesions in those cells. We then measured the adhesion forces of the cells and found that the MiuA treated ones exert a lower adhesion force and energy. Additionally, we placed cells on a patterned soft substrate, which they deform by contracting. Measuring these deformations, we were able to calculate the contractile energy of single cells. We showed that MiuA does not affect the contractile energy of cells. Lastly, we conducted 1D migration experiments with RPE-1 cells and neutrophils confined in PDMS microchannels. MiuA did not affect the amoeboid migration behavior in either cell line.

Taken together, our findings suggest that the effect of MiuA on the length of actin filaments and focal adhesions is important in mesenchymal migration but not in amoeboid migration. This is because MiuA influences mechanisms that are key to adhesion-based migration but not essential to amoeboid migration.

## Materials and methods

### Cell culture

We cultured RPE-1 cells, wildtype and transfected with the LifeAct mCherry marker (as described by Maiuri et al. in 2015 [1]), and MEF cells transfected with GFP-vinculin at 37 °C and 5% $CO_2$ in Dulbecco's Modified Eagle Medium Nutrient Mix F12 with 10% FBS, 1% GlutaMax, and 1% Penicillin Streptomycin (Thermo Fisher). The RPE-1 cells were kindly gifted by the lab of Mathieu Piel, Institute Curie, Paris, while the MEF GFP vinculin cells were kindly given by Jennifer Kasper, Leibniz-Institute for New Materials. We cultured HL60 cells (bought from ATCC) at 37 °C and 5% $CO_2$ in RPMI 1640 Medium (Thermo Fisher) with 10% FBS, 1% GlutaMax, and 1% Penicillin Streptomycin (Thermo Fisher). We differentiated the HL60 cells by adding 1.3% of DMSO to the medium and incubating them for two days. A list of the cells we used in this study can be seen in Table 1.

### Phalloidin staining

We allowed the MEF GFP vinculin cells and REP-1 wt cells to fully spread on a fibronectin-coated glass surface (fibronectin concentration 25 µg/ml). Then, we treated them with DMSO (control) and 5 nM of MiuA for 1 hour before we fixed them with 4% PFA, dissolved their membrane using a 0.1% Triton X100 solution, and washed them three times with PBS. We added a 3% BSA solution to block unspecific binding sites. After one hour, we added a 1:1000 solution of Phalloidin-iFluor 555 (abcam: ab176756) and 3% BSA for another hour. We then washed the cells three times with PBS and mounted them with Fluoromount G+DAPI.

**Table 1. List of the cells used in this study.**

| Cell line | Info | Provider | Usage |
|---|---|---|---|
| RPE-1 LifeAct mCherry | hTERT RPE-1 cells transfected with LifeAct mCherry | Lab of Matthieu Piel (Paris) | Migration assay (channels), Trypsin assay, Focal adhesion staining, FluidFM measurements |
| RPE-1 wt | hTERT RPE-1 cells (wildtype) | Lab of Matthieu Piel (Paris) | Contractility screening, dosage dependency of MiuA |
| MEF GFP vinculin | Transfected with GFP vinculin | Dr. Jennifer Kasper, Leibniz-Institute of new materials (Saarbrücken) | Focal adhesion staining, Phalloidin staining |
| Neutrophils | HL60 cells differentiated into neutrophils | HL60 cells bought from ATCC | Migration assay (channels), Focal adhesion staining |

## Focal adhesion staining

We took fully spread and adhered RPE-1 LifeAct mCherry cells on fibronectin (25 µg/ml) coated glass and treated them for 1 hour with DMSO (control) or 20 nM MiuA before we fixed them with 4% PFA and dissolved their membrane. Additionally, we treated the RPE-1 LifeAct mCherry cells and the neutrophils with DMSO (control) or 20 nM MiuA before we placed them inside PDMS microchannels (coated with 25 µg/ml fibronectin or 100 µg/ml pll-g-PEG) and let them enter the channels. After at least 4 hours, we fixed those cells too with 4% PFA and dissolved their membrane. For this, we kept the cells for 10 minutes in a 0.1% Triton X100 solution and washed them afterwards three times with PBS. We added a 3% BSA solution to block unspecific binding sites. After one hour, we added a 1:1000 solution of paxillin polyclonal antibodies (ThermoFisher PA-34910) and 3% BSA. After another hour, we washed the cells three times with PBS and mounted them with Fluoromount G+DAPI. For focal adhesions in MEF GFP vinculin cells, we seeded them on a glass coverslip coated with fibronectin (25 µg/ml) and put them in the incubator for at least 4 hours. Once the cells were fully adhered, we treated them with MiuA (5 nM) or DMSO for 1 hour before we fixed them with 4% PFA and mounted them with Fluoromount G+DAPI. We repeated all experiments three times.

## Trypsin assay

We seeded RPE-1 LifeAct mCherry cells on a glass coverslip coated with fibronectin (25 µg/ml) and allowed them to fully adhere. After at least four hours of spreading time, we treated the cells for one hour with DMSO (control) or 20 nM of MiuA. We placed the cells inside a microscope with temperature and $CO_2$ control before we added trypsin and imaged the cells for 5 minutes. We took images every 20 seconds. We repeated the experiment three times.

## Single-cell force spectroscopy

To measure the adhesion forces of RPE-1LifeAct mCherry cells to fibronectin, we treated tissue culture dishes (TPP, Trasadingen, Switzerland) with plasma for 3 minutes, coated them with fibronectin (25 µg/ml) and incubated them for 1 hour at room temperature. After the preparation of the culture dishes, we added RPE-1 LifeAct mCherry cells ($10^6$ cells/ dish) along with the compound of interest (DMSO or MiuA). We allowed the cells to adhere to the fibronectin for 4 hours in an incubator (37 °C, 5% $CO_2$). To perform single-cell force spectroscopy outside of a $CO_2$ incubator, we added 25 µl of HEPES per 1 ml of cell medium to the cell medium.

We performed single-cell force measurements at 37 °C using a Nanowizard IV XP AFM with a CellHesion 200 Head (Bruker-JPK, Santa Barbara, CA, USA), a FluidFM microfluid control system V2 Platinum (Cytosurge, Glattburg, Switzerland), and a JPK PetriDishHeater. We approached (setpoint 8nN) RPE-1 LifeAct mCherry cells with a Cytosurge FluidFM micropipette (aperture 4 µm, stiffness 2 N/m) and immobilized the cells to the micropipette with a vacuum (−500 mbar) while maintaining the micropipette at a constant height or force. We then detached the cells from the fibronectin by

retracting the micropipette (z-length 50 µm, z-velocity 0.8 µm/s). We used an inverted-light microscope (Zeiss AG, Oberkochen, Germany) to observe the detachment process. We analyzed the force–distance curves using JPK Data Processing Software, version 7.0.128. We repeated the experiment three times.

**Contractility screening**

We prepared micropatterned glass coverslips following the protocol used by Azioune et al. [31]. First, we cleaned the glass coverslips with ethanol prior to a plasma treatment lasting 3 minutes. Then, we coated the coverslip with a 50 µl droplet of a 100 µg/ml pll-g-PEG solution for 1 hour at room temperature. We washed the pegylated coverslips in pure water and placed them on an activated quartz-chromium photomask (ROSE Fotomasken, Germany) using a 4.5 µl droplet of pure water. After 6 minutes of deep-UV irradiation (200 nm), we lifted the glass coverslips from the photomask and covered them with a 50 µl droplet of a 25 µg/ml solution of fibronectin mixed with 0.1% BSA+Alexa555 (Thermo Fisher).

After at least 30 minutes, we prepared a solution of 200 µl acrylamide, 96 µl bis-acrylamide, and 198.5 µl PBS; vortexed it; and degassed it with nitrogen for a minimum of 2 minutes. To this, we added 5 µl of a 10% APS (ammonium persulphate) solution and 0.5 µl TEMED (Tetramethylethylenediamine), before mixing it with a pipette. Afterwards, we pipetted 50 µl droplets on the patterned glass coverslips and placed an acryl-silanized coverslip on top. We waited at least 15 minutes before we immersed the coverslips in PBS for a minimum of 1 hour to allow the hydrogel to swell. After this, we removed the top coverslip, which was attached to the gel and the micropatterns. We seeded 100,000 RPE-1 wildtype cells on each coverslip and placed them in the incubator overnight. The next day, we treated them for 1 hour with DMSO (control) or 20 nM of MiuA before we fixed the cells with 4% PFA. We repeated each experiment three times.

For the analysis, we measured the change in the micropattern area and calculated the contractile energy using the following formula (derived from Ghagre et al. [30] and Landau and Lifshitz *Theory of Elasticity,* [32,33]):

$$E_{contractile} = \frac{E(1-\sigma)}{(1+\sigma)\left(\left(2(1-\sigma)+\sigma\right)^2 - \sigma^2\right)\cdot 3\cdot\pi^{\frac{3}{2}}}\left(A_i^{\frac{3}{2}} - A_f^{\frac{3}{2}}\right)$$

where

- E is the Young's modulus,
- σ is the Poisson ratio (here 0.5 for an incompressible material),
- $A_i$ is the initial pattern area and
- $A_f$ is the final pattern area.

Our formula differs from the one used by Ghagre et al*.,* as we reduced the necessary parameters from two to one by solving the following equilibrium equation for an elastic deformation of a substrate in the xy-plane:

$$(1-2\sigma)\Delta\vec{u} + \vec{\nabla}\left(\vec{\nabla}\,\vec{u}\right) = \vec{0}$$

where we used polar coordinates to describe our displacement vector $\vec{u}$ using only radial displacement.

$$\vec{u} = (r_f - r_i)\,\vec{e}_r$$

We present the full calculations in the SI.

## Microfabrication of PDMS microchannels

We mixed the RTV615 A + B compound (Momentive) at a ratio of 10:1 and proceeded as described by Vesperini et al. [34]. The microchannels had a diameter of 10x10 μm for the RPE-1 LifeAct mCherry cells and 5x5 μm for the neutrophils. We treated all the cells with 250 ng/ml Hoechst prior to imaging. We coated the PDMS channels with 100 μg/ml poly-L-lysine-grafted poly-ethylene-glycol (=pll-g-PEG) (for the migration assays and focal adhesion staining) or 25 μg/ml of fibronectin (for focal adhesion staining) and washed them three times with PBS before we completely submerged them in the cell medium. We kept them for at least one hour in the cell medium before we added the cells. For the experiments, we added around 100,000 cells to one of the loading channels. These cells were already suspended in cell medium containing only DMSO (control) or 20 nM of MiuA. We repeated each experiment three times.

## Microscopy

We imaged the fixed samples using a ZEISS Axio observer (base version: epifluorescence, non-confocal, Zeiss AG, Oberkochen, Germany) with a 63x magnification oil objective. We took time-lapse images of migrating cells with a Nikon Eclipse Ti microscope using a 10x magnification objective. We set the temperature inside the microscope's incubation chamber to 37 °C and kept the $CO_2$ level at 5%. To ensure that there were no fluctuations in the settings, we allowed the system to settle for one hour prior to the start of the experiments. We observed cell movement by treating living cells with 250 ng/ml Hoechst for 30 minutes before the start of the recording. We took pictures every five minutes for at least ten hours.

## Image analysis

We performed image analysis with the open-source software Fiji (ImageJ) [35]. We analyzed time-lapse images of migrating cells using the plug-in TrackMate [36,37]. We determined the number and size of the focal adhesions via paxillin or vinculin staining. We used a threshold for the fluorescence signal and analyzed the images with Fiji's "Analyze particles" function.

## Statistical analysis

We conducted student's t tests on all experimental data using a home-built Python script (t-test_ind-function from SciPy 1.14.1). We calculated the p values and assigned significance as follows: no significance (n.s.) $p > 0.05$, * $p < 0.05$, ** $p < 0.01$, and *** $p < 0.001$.

We include all exact p values in the supplementary information S1 File.

## English correction

During the preparation of this work, we used ChatGPT-3 (OpenAI, 2021) and DeepL (DeepL writer, Beta, 2024) to improve the language of the manuscript. After using these tools, we reviewed and edited the content as needed and hereby take responsibility for the content of the publication.

# Results and discussion

## Effects of miuraenamide A on focal adhesions

The actin stabilizer miuraenamide A was first identified in 2006 by Iizuka et al. [21] and subsequently synthesized by Karmann et al. in 2015 [20]. It can enter living cells and stabilizes actin filaments by modifying the conformation of actin monomers, thereby preventing cofilin, an actin-severing protein, from accessing its binding site [17].

In this work, we first tested whether the actin stabilizing effect of MiuA applies to different cell lines.

In our previous work [23], we already reported the actin stabilizing effect in RPE-1 cells as well as the reduction of 2D migration in RPE-1 and MEF cells when treated with MiuA. However, we did not look at the effect of MiuA on the actin

filament length in MEF. We have therefore now done this by seeding MEF GFP vinculin cells on glass cover slips coated with fibronectin (25 µg/ml) and allowing them to fully adhere. Fig 1a and S1 Fig shows an example of fully adhered spread cells in a side-by-side comparison between RPE-1 and MEF cells. We found that the MEF cells treated with MiuA did not increase their spreading area (Fig 1b), remaining at around 550 µm$^2$, while the average length of their actin filaments increased from 10.2 µm in untreated cells to 19.1 µm in MiuA treated cells (Fig 1c). This effect partially aligns with the effects in RPE-1 cells that we already reported [23]. In both cell lines the average length of actin filaments increased but only RPE-1 cells showed an additional increase in spreading area. We expected to see an increase in the spreading area of MEF cells as well, since Gegenfurtner et al. [22] reported that MiuA positively influences genes related to cell spreading. We hypothesize that this effect in our case only takes place in RPE-1 cells and not in MEF cells since the concentration of MiuA was higher in RPE-1 cells (20 nM) compared to MEF cells (5 nM). When we tested different concentrations, we found 20 nM to be cytotoxic for MEF cells and therefore reduced to concentration of MiuA accordingly. Additionally, when we measured the aspect ratio and circularity of RPE-1 and MEF cells we found no significant differences (Fig 1d, 1e, 1g) or a low significance (Fig 1f, p-value ~ 0.04). The results align with our expectations. We found RPE-1 cells increased their spreading area but initially exhibited a uniform distribution of actin filaments. As all cells were treated post-adherence, we did not anticipate MiuA to alter this distribution or cell geometry. Conversely, we found MEF cells did not increase their spreading area. We propose that in MEF cells MiuA only stabilized pre-existing actin filaments and elongated within the cell's established borders, leaving the cell geometry unaffected. Treating RPE-1 cells with a lower concentration of MiuA (5 nM) resulted in longer actin filaments and an increased spreading area, but not significantly (S2 Fig). However, we conclude that the treatment with MiuA leads to an elongation and stabilization of actin filaments within living cells regardless of the cell line.

Linked to actin filaments are focal adhesions, which connect the cytoskeleton and the extracellular matrix. Focal adhesions are clusters of proteins, including integrins, paxillin, vinculin, and talin [24–29]. These clusters form the connection between the outer extracellular matrix and the inner actin cytoskeleton. Although it is established that the size of focal adhesions affects the composition of the stress fibers connected to them [38], the relationship between actin filament length and focal adhesions remains largely unexplored. We therefore expected that targeting actin with MiuA would have an indirect effect on focal adhesions.

In this study we investigated the correlation between actin filament length and focal adhesion number and size. We seeded RPE-1 LifeAct mCherry and MEF GFP vinculin cells on a fibronectin-coated glass surface, treated them with MiuA, and fixed them after they had fully spread (Fig 2a-2e). Our analysis of their focal adhesions showed that the average size of focal adhesions in cells treated with MiuA was smaller than in the control group (Fig 2f, 2i). With MiuA treatment the average focal adhesion size reduced from 1.258 µm$^2$ to 0.977 µm$^2$ in RPE-1 cells and from 0.839 µm$^2$ to 0.603 µm$^2$ in MEF cells. However, the average number of focal adhesions increased significantly in RPE-1 cells (Fig 2g) from 57.8 to 103.2, and in MEF cells (Fig 2j) from 21.5 to 39.1 after treatment with MiuA. This resulted in a larger total area of focal adhesions in RPE-1 cells but not in MEF cells (Fig 2h, 2k). Unlike in RPE-1 cells, MiuA treatment of MEF cells did not change their spreading area and we therefore conclude that the total focal adhesion area normalized to the spreading area of the cells did not change.

Since focal adhesions are crucial for the overall adhesion of cells, we next measured how cell adhesion differs depending on the size of their focal adhesions.

## Adhesion forces

Knowing that cells with stabilized and elongated actin filaments express more but smaller focal adhesions, we next quantified the relationship between focal adhesion size and the adhesive force exerted by the cell. Previous studies on single focal adhesions have shown that there is a positive correlation between the size and the force that cells can exert on the underlying matrix [38–43]. However, the effect of average focal adhesion size on the adhesion properties of cells as a whole remains unclear.

 

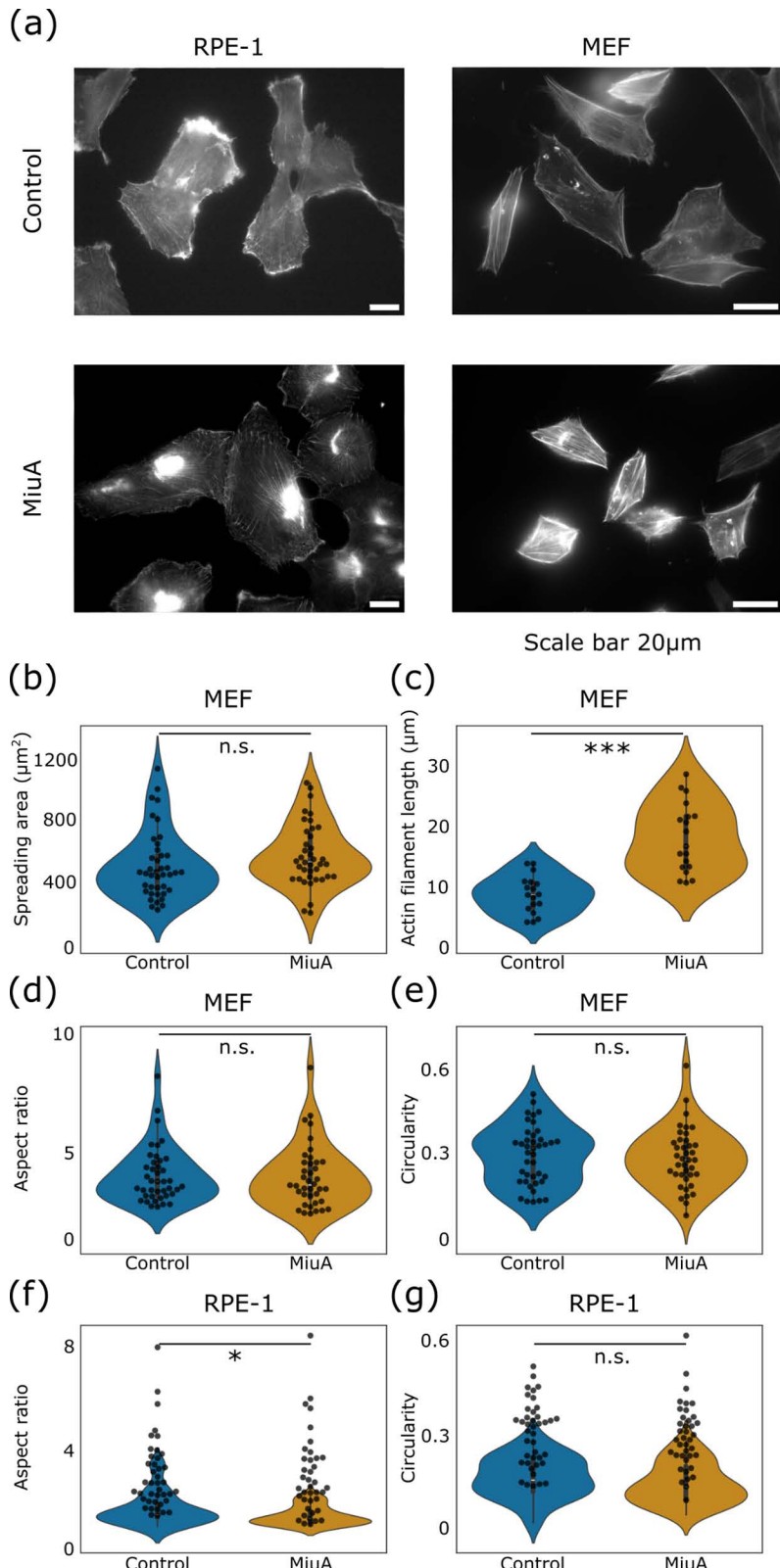

**Fig 1. Effect of MiuA on actin. (a)** Representative images of the actin cytoskeleton in RPE-1 LifeAct mCherry and MEF GFP vinculin cells when treated with MiuA. Top: control cells. Bottom: MiuA treated cells (20 nM for RPE-1 LifeAct mCherry and 5 nM for MEF GFP vinculin). Actin filaments in MEF GFP

vinculin cells were stained with Phalloidin-iFluor 555. Scale bar 20 μm. **(b)** Spreading area of MEF GFP vinculin cells seeded on a fibronectin-coated glass surface; untreated (control) cells and cells treated with 5 nM MiuA. **(c)** Mean length of actin filaments in MEF GFP vinculin cells, spread on fibronectin-coated glass surface. Effect of MiuA on the aspect Ratio of MEF **(d)** and RPE-1 **(f)** cells. Effects of MiuA on the circularity of MEF **(e)** and RPE-1 **(g)** cells. Number of cells: MEF: Control: 41, MiuA: 38; RPE-1: Control: 38, MiuA: 35. A statistical analysis was performed using Student's t test. n.s.: $p > 0.05$; *: $p < 0.05$; **: $p < 0.01$; and ***: $p < 0.005$.

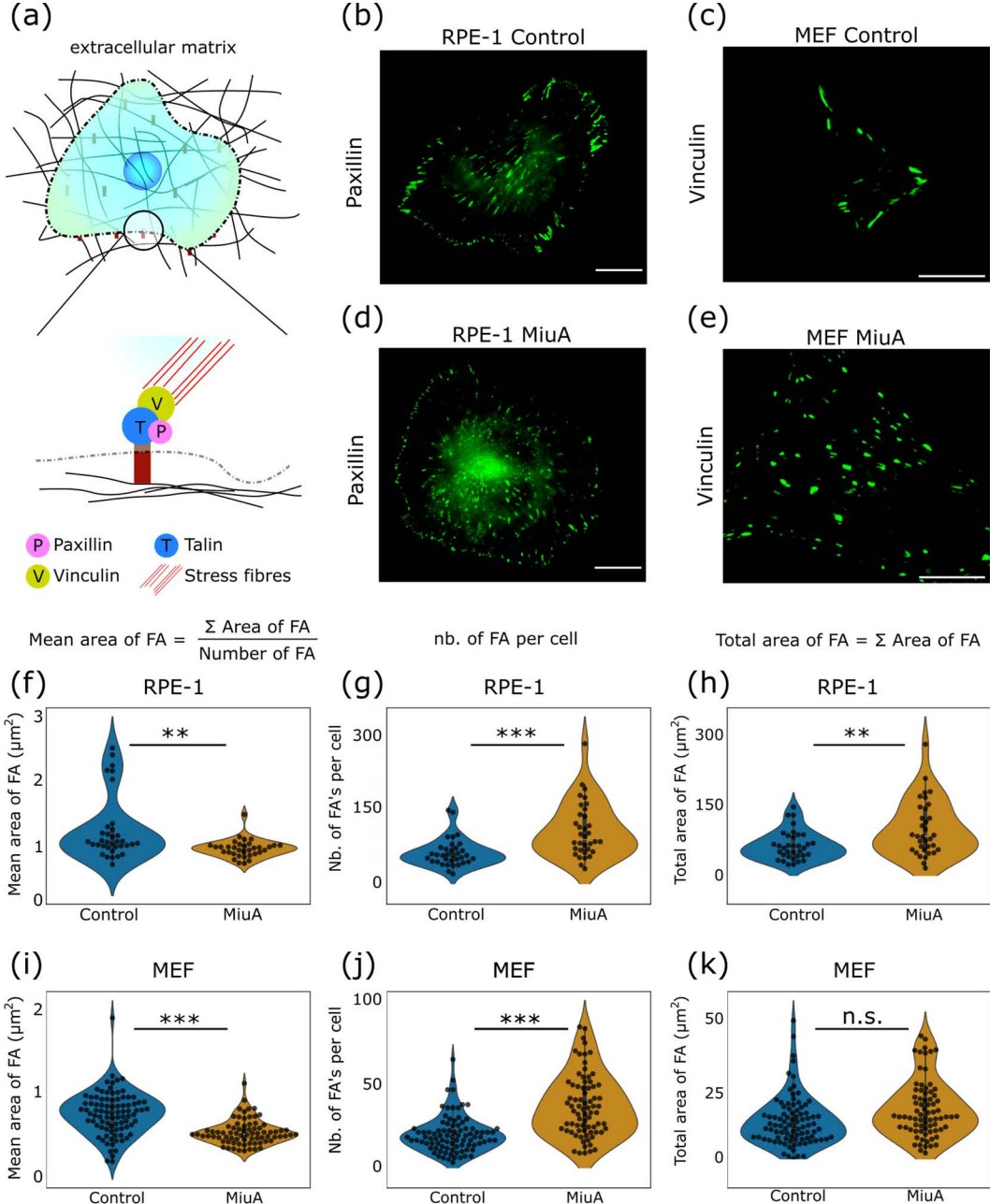

**Fig 2. Effects of MiuA on focal adhesions (FAs). (a)** Schematic of a cell adhering to a fibronectin-coated surface. Epifluorescence images of focal adhesions in RPE-1 LifeAct mCherry cells with paxillin stained **(b)(d)** as well as MEF GFP vinculin cells **(c)(e)**. Mean and total values of FA size and their number for RPE-1 cells **(f)-(h)** and MEF cells **(i)-(k)**. In both cases, the number and mean area of focal adhesions per cell increased when using MiuA. Data on the number of focal adhesions taken from Baltes et al. [23]. A statistical analysis was performed using Student's t test. n.s.: $p > 0.05$; *: $p < 0.05$; **: $p < 0.01$; and ***: $p < 0.005$. Scale bar: 15 μm. Number of cells: RPE-1, 35 control and 34 MiuA; MEF, 90 control and 76 MiuA.

In order to measure the adhesion forces of cells, we performed single-cell force spectroscopy, a method well suited to measure the force required to detach a single-cell from the substrate to which it adheres [44–46,55–57]. We collected force-distance curves, allowing us to calculate adhesion parameters such as adhesion energy, force and rupture length (Fig 3a). We seeded RPE-1 cells in a plastic bottom dish coated with fibronectin (25 µg/ml). We allowed the cells to fully spread before we measured force-distance curves. We found that the basic form of the force-distance curves remained unchanged between control and MiuA treated cells, but observed significant differences in adhesion force and energy (Fig 3b,3c). From our measurements, it is evident that the addition of MiuA significantly reduced the adhesion energy of RPE-1 cells (Fig 3d), specifically the average dropped from 3.36 pJ in untreated cells to 1.06 pJ in MiuA treated cells. The average adhesion force showed the same trend (Fig 3e), such that untreated cells exhibited an average adhesion force of 0.279 µN while MiuA treatment reduced this to 0.074 µN.

Despite the elongation of actin filaments in MiuA treated cells, the detachment length did not appear to increase, remaining at around 31 µm for both untreated and MiuA treated cells (Fig 3f). However, the MiuA treated cells displayed a greater spread in detachment length, which could indicate increased variation in the elongated actin filaments.

Interestingly, when we used trypsin to detach the cells, we saw that RPE-1 cells treated with MiuA took longer than the untreated cells (S3 Fig). This is contrary to the expectations from our single-cell force spectroscopy measurements. We suggest that this is due to the different mechanisms used by the two approaches. Trypsin detachment of cells is achieved by chemical degradation of adhesion sites in contrast to single-cell force spectroscopy which physically detaches cells. Since trypsin is a digestive enzyme that can degrade proteins such as integrins in focal adhesions [47], the time it takes for cells to detach after the addition of trypsin correlates with the time required for protein degradation. We propose that, due to the increased number of focal adhesion sites in MiuA-treated cells, the degradation process takes longer.

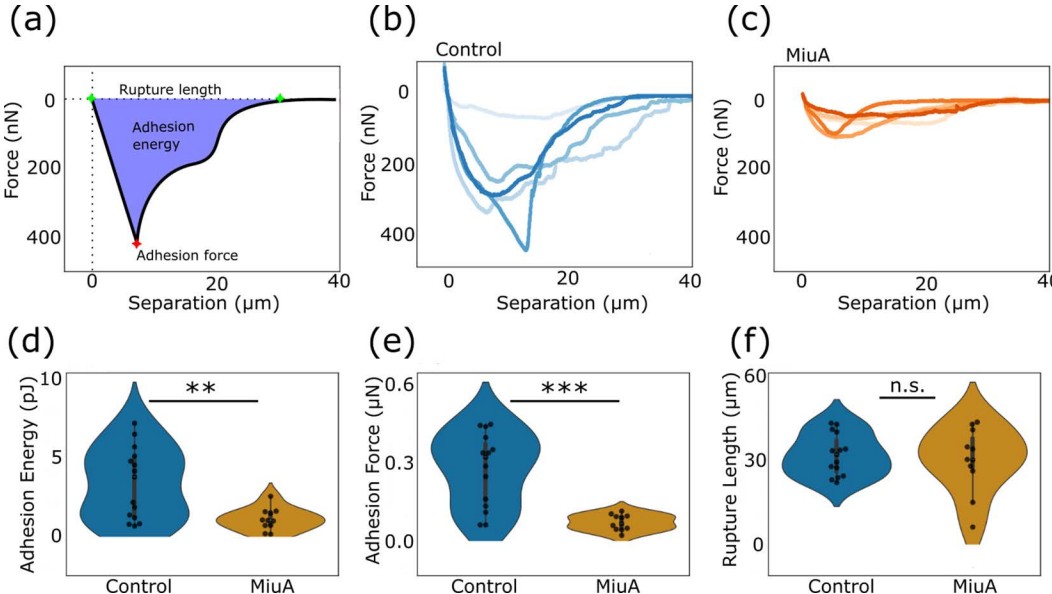

**Fig 3. Fluid force measurements (FluidFM) on RPE-1 cells. (a)** Schematic sketch of the force–distance curve and explanation of adhesion energy, adhesion force, and rupture length. Representative force–distance curves for the control (b) and MiuA-treated cells **(c)**. The violin plots of adhesion energy **(d)**, adhesion force **(e)**, and rupture length (f) show that energy and force are reduced in cells with smaller FAs, while rupture length remains unchanged. A statistical analysis was performed using Student's t test. n.s.: p > 0.05; *: p < 0.05; **: p < 0.01; and ***: p < 0.005. Number of cells: 15 control and 11 MiuA.

Taken together, neither the total number nor the total area of focal adhesions appear to have the most significant effect on the adhesion force and energy. Instead, the reduction in the size of the individual focal adhesions appears to have the most significant effect in reducing the adhesion force and energy.

## Cell contractility

In addition to actin polymerization and cellular adhesion, cells require contraction in all types of migration [47–51]. Therefore, in this work we also investigated the influence of MiuA on cell contractility.

We performed a pattern-based contractility screening assay (PaCS) [32] in which RPE-1 cells are placed on a polyacrylamide gel (Fig 4a) with a stiffness of 10 kPa. Treating the surface of these gels with fibronectin allows the cells to adhere only to a specific fluorescently labeled area, known as a micropattern [31]. These micropatterns are small enough that when a cell contracts, the underlying micropattern and substrate equally deform. By monitoring this deformation, the quantification of changes in the pattern area can be detected, thus calculating the contractile energy exerted by the cell to deform the pattern.

Interestingly, we found that cells treated with MiuA did not show significant changes in contractile energy compared to untreated cells (Fig 4b). For untreated cells the average contractile energy was 4.907 pJ, while being decreased to 4.735 pJ in MiuA treated cells. To verify the sensitivity of our assay, we also treated cells with 10 μM blebbistatin, which inhibits the motor protein myosin II. As expected, the treatment with blebbistatin reduced the contractile energy of RPE-1 cells (Fig 4c-4e), dropping from approximately 4.80 pJ in untreated and MiuA treated cells to 2.61 pJ in blebbistatin treated cells. This gives us confidence that our PaCS assay is sensitive enough to detect changes in contractility and that MiuA did not alter this contractility.

Although MiuA reportedly changes the architecture of the actin network in cells, which has an impact on cellular mechanics such as adhesion, we do not know of any reports about the effects of MiuA on the interaction between actin and myosin motor proteins. Since MiuA does not disrupt the actin network and presumably does not block myosin from

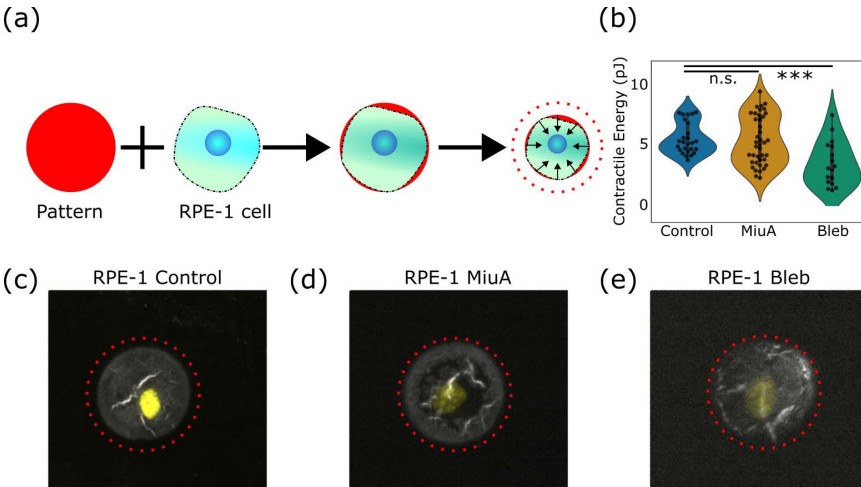

**Fig 4. Contractile energy of RPE-1 cells on poly-acrylamide gels. (a)** Schematic diagram of the technique, where a cell adheres to a pattern on a soft gel and deforms it, with the red dotted lines representing the initial pattern outline. **(c)**, **(d)**, **(e)** Representation of a single cell (nucleus in yellow on a soft gel pattern (grey) showing the area before deformation as a dashed circle (red). When treated with MiuA cells do not show any significant change in contractile energy (b) or pattern area after deformation, whereas treatment with blebbistatin reduces contractile energy (e) and increases pattern area after deformation compared to control. Scale bar: 20 μm. Statistical analysis was performed using Student's t test. n.s.: $p > 0.05$, *: $p < 0.05$, **$p < 0.01$, ***: $p < 0.005$. Number of cells: Control: 38, MiuA: 40, blebbistatin: 22.

actin filaments, we assume that the interaction between the two is unchanged in our experiments. This explains why we observe no changes in contractility in our experiments. However, further studies on the interactions between actin, MiuA and motor proteins, especially on co-localization are required to fully answer this question.

We next tested how amoeboid migration was affected by MiuA.

## Cell migration

Cells can migrate using a non-adhesive process known as amoeboid migration, which does not rely on adhesion but on friction and contractility [1,3,52–54]. RPE-1 cells are known to be able to switch between migration modes, depending on the environment, and neutrophils are widely used as a model system for amoeboid migration [1,2]. To investigate the effects of MiuA treatment on amoeboid migration, we used both cell lines to avoid specific effects linked to the respective cell line. We hypothesized that amoeboid migrating cells would not be affected by actin filament stabilization as the actin stabilizer MiuA does not change their contractility. To test this, we forced RPE-1 cells to migrate in an amoeboid mode [1] by placing RPE-1 cells in confining microchannels and recording their migration speed and persistence (Fig 5a-5d). To

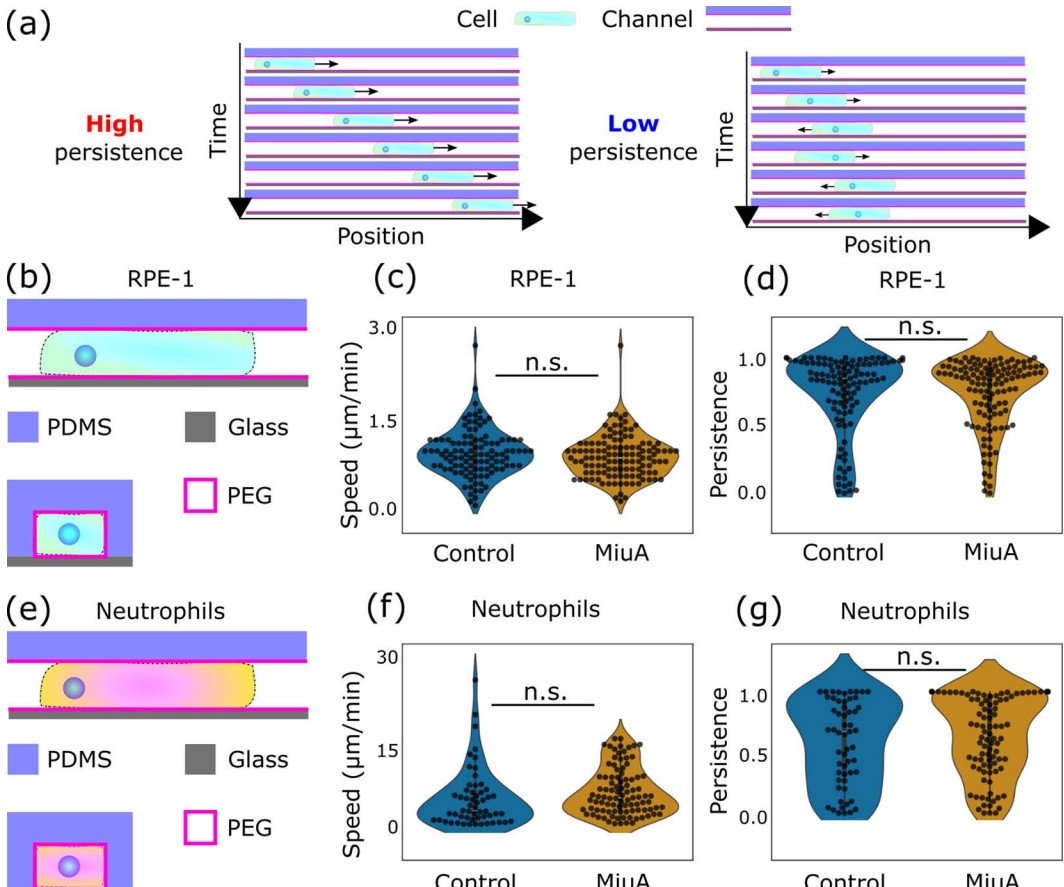

**Fig 5. Effects of MiuA on migrating cells. (a)** Visual representation of "persistence". A high persistence means little directional changes, whereas a low persistence indicates a cell that changes its direction a lot. Schematic representation of amoeboid migrating cells **(b, e)**. Under confinement, RPE-1 LifeAct mCherry cells show no differences in speed and persistence **(c, d)**. The same results occur when neutrophils migrate in confinement **(f, g)**. A statistical analysis was performed using Student's t test. n.s.: p > 0.05; *: p < 0.05; **: p < 0.01; and ***: p < 0.005. Number of cells: RPE-1: 109 control and 119 MiuA. Neutrophils: 56 control and 102 MiuA.

ensure that the observed migration mode was amoeboid, we coated the PDMS microchannels with pll-g-PEG, which is known to hinder adhesion. After seeding the cells and staining their focal adhesions, we saw that the cells inside pll-g-PEG-coated channels did indeed not express focal adhesions, unlike cells within fibronectin coated channels (S4 Fig). As hypothesized, we did not find significant changes in the migration of MiuA treated RPE-1 cells in pll-g-PEG-coated channels. We measured speeds of 0.918 µm/min and 0.858 µm/min for untreated cells and MiuA treated cells respectively, while the persistence in both cases was 0.740. By "persistence" in this study we refer to the tendency of cells to change their direction while migrating. The value of this persistence lies between 0 (constantly changing direction) and 1 (no changes in direction). Additionally, we saw a positive correlation between the speed and persistence in RPE-1 cells, meaning that faster RPE-1 cells are also more persistent in their movement.

To confirm that this behavior is general to amoeboid movement and not just a property specific to RPE-1 cells, we repeated the same migration experiment in pll-g-PEG-coated channels using neutrophils (Fig 5e-5g). Like RPE-1 cells, neutrophils did not change their migration speed or persistence after the treatment with MiuA significantly. Their average speed was 4.172 µm/min in untreated cells and 5.363 µm/min in MiuA treated cells, and their persistence was 0.577 for untreated cells and 0.636 for MiuA treated ones. Although the correlation between speed and persistence was not as high as in RPE-1 cells, neutrophils still showed the same trend in which faster cells tend to migrate in a more persistent manner. Taken together, we conclude that the actin filament stabilization effect of MiuA has no significant effect on friction-based migration, confirming our hypothesis.

## Conclusion

In this study we investigated the interplay between actin filament length, focal adhesions, adhesion forces, adhesion energy, and cell migration and explained that they are all interrelated.

First, we confirmed the actin filament stabilizing effects of MiuA in MEF cells and showed that the treatment does indeed elongate actin filaments in MEF cells without affecting the spreading area. This confirmed our hypothesis that MiuA stabilizes and elongates actin filaments in living cells regardless of the used cell line in this study.

Second, we tested how this stabilization of actin affects the size and number of focal adhesions and found an increase in the number of focal adhesions while reducing the size of individual focal adhesions. We also tested how the size and number of focal adhesions affect the adhesion properties of the cells. We found that RPE-1 cells treated with MiuA had more but smaller focal adhesions and adhered less, while their rupture length was unaffected.

Furthermore, our contractility measurements on RPE-1 cells revealed that MiuA does not affect the contractility.

We hypothesized that the actin filament stabilization would only affect adhesion-based migration (mesenchymal migration), since focal adhesions and adhesion properties were altered while contractility remained unaffected. Therefore, we expected that amoeboid migration, which does not rely on adhesion, would not be altered.

We have now confirmed this hypothesis and also explained our previous findings about the reduction of mesenchymal 2D migration of RPE-1 and MEF cells (S5 Fig) [23]. In our previous publication [23], we hypothesized that this was due to an increase in adhesion force. However, in this work we have now disproved this hypothesis by measuring a reduction of adhesion force and energy. Instead, we now conclude that the treatment with MiuA reduces adhesion properties, which impedes the cell from exerting the forces needed to perform mesenchymal migration.

On the other hand, when we placed RPE-1 cells and neutrophils in pll-g-PEG-coated microchannels and forced them to migrate in an amoeboid mode, neither their speed nor their persistence changed after the treatment with MiuA. This confirmed our hypothesis that non-adhesion-based migration would not be affected by MiuA.

Taken together, we conclude that MiuA treatment affects adhesion-based migration, but not amoeboid migration in cells.

We showed in this work that MiuA can be used as a reliable tool to alter cellular adhesion properties in cells while leaving cellular contractility unaffected. Further experiments using this tool will lead to a better understanding of the different mechanisms of migration in immune and cancer cells.

## Supporting information

**S1 Fig.  Additional images of MEF cells.** Left: control, right: 5nM MiuA. Scale bar 20μm.
(TIF)

**S2 Fig.  Dosage dependent effect of MiuA on RPE1 cells.** (a) Spreading area of RPE-1 wt cells on a fibronectin (25 μg/ml) coated glass surface. The treatment with 20nM of MiuA significantly increased the spreading area, whereas 5nM showed no significant effect. (b) Mean actin filament length per cell in RPE-1 wt cells. Treatment with 20nM MiuA significantly increased the mean actin filament length, while 5nM showed no significant effect. Number of cells: 35 (a), 30 (b). A statistical analysis was performed using Student's t test. n.s.: $p > 0.05$; *: $p < 0.05$; **: $p < 0.01$; and ***: $p < 0.005$.
(TIF)

**S3 Fig.  Trypsin assay for RPE-1 LifeAct mCherry cells.** Fully spread RPE-1 LifeAct mCherry cells were trypsinized while being monitored with a video microscope. Pictures were taken every 20 seconds where the time stamp (yellow label in the upper left corner) denotes the time after adding trypsin. The first picture "0 s" was taken immediately after trypsin was added. Top: Control cells treated with DMSO. Bottom: MiuA (20nM) treated cells. Scale bar 20 μm.
(TIF)

**S4 Fig.  Focal adhesion staining in migrating RPE-1 LifeAct mCherry cells and neutrophils.** From left to right: Paxillin immunostaining with zoomed in area (red square) in green. DAPI staining of the nucleus in yellow. Brightfield images of RPE-1 cells and neutrophils. Composite images of the paxillin, nucleus and brightfield images. From Top to bottom: (a) RPE-1 cells on 2D fibronectin (25 μg/ml) coated surface. (b) RPE-1 cells inside 10x10 μm PDMS microchannels with fibronectin coating. (c) RPE-1 cells on a PEG-coated surface. (d) RPE-1 cells in PEG-coated 10x10 μm PDMS microchannels. (e) Neutrophils on a PEG-coated 2D surface. (f) Neutrophils inside a PEG-coated 5x5 μm PDMS microchannel. Scale bars: 20 μm.
(TIF)

**S5 Fig.  Effects of MiuA on mesenchymal migrating RPE-1 cells.** Schematic representation of a mesenchymal migrating cell on a fibronectin line (a). On these fibronectin lines, RPE-1 LifeAct mCherry cells treated with 20nM MiuA reduced their speed (b) and persistence (c). A statistical analysis was performed using Student's t test: n.s.: $p > 0.05$; *: $p < 0.05$; **: $p < 0.01$; and ***: $p < 0.005$. Number of cells: 55 control and 77 MiuA. Data replotted from Baltes et al. [23].
(TIF)

**S1 File.  P-values Excel-file with all our p-values derived from the statistical tests performed on the data.**
(XLSX)

**S2 File.  Related article PDF-file of the related article mentioned in the text.** This article refers to reference Baltes et al. [23].
(PDF)

**S3 File.  Calculation of contractile energy PDF-file containing a detailed description of the calculation of the contractile energy in cells on an elastic surface.**
(PDF)

## Acknowledgments

We thank Ajinkya Ghagre and Allen Ehrlicher for their advice on the soft gel patterning technique PaCS, as well as Matthieu Piel and Jennifer Kasper for providing us with the cell lines.

## Author contributions

**Conceptualization:** Carsten Alexander Baltes, Franziska Lautenschläger.

**Data curation:** Carsten Alexander Baltes, Friederike Nolle, Kathi Michèle Kaiser, Erbara Gjana, Kristin Sander.

**Formal analysis:** Carsten Alexander Baltes, Friederike Nolle, Kathi Michèle Kaiser.

**Funding acquisition:** Karin Jacobs, Franziska Lautenschläger.

**Investigation:** Carsten Alexander Baltes, Franziska Lautenschläger.

**Methodology:** Carsten Alexander Baltes, Friederike Nolle, Kathi Michèle Kaiser, Erbara Gjana, Rhoda Joy Hawkins.

**Project administration:** Franziska Lautenschläger.

**Resources:** Franziska Lautenschläger.

**Supervision:** Karin Jacobs, Rhoda Joy Hawkins, Franziska Lautenschläger.

**Validation:** Carsten Alexander Baltes.

**Visualization:** Carsten Alexander Baltes, Kathi Michèle Kaiser, Erbara Gjana, Kristin Sander.

**Writing – original draft:** Carsten Alexander Baltes.

**Writing – review & editing:** Carsten Alexander Baltes, Friederike Nolle, Karin Jacobs, Franziska Lautenschläger.

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
