## [Decision Letter · Decision Letter 0]

7 Apr 2025

Dear Dr. Lautenschläger,

Thank you for submitting your manuscript to PLOS ONE. After careful consideration, we feel that it has merit but does not fully meet PLOS ONE’s publication criteria as it currently stands. Therefore, we invite you to submit a revised version of the manuscript that addresses the points raised during the review process.

We look forward to receiving your revised manuscript.

Kind regards,

Florian Rehfeldt

Academic Editor

PLOS ONE

**Journal Requirements:**

1. When submitting your revision, we need you to address these additional requirements. Please ensure that your manuscript meets PLOS ONE's style requirements, including those for file naming. The PLOS ONE style templates can be found at https://journals.plos.org/plosone/s/file?id=wjVg/PLOSOne_formatting_sample_main_body.pdf and https://journals.plos.org/plosone/s/file?id=ba62/PLOSOne_formatting_sample_title_authors_affiliations.pdf 2. Please note that PLOS ONE has specific guidelines on code sharing for submissions in which author-generated code underpins the findings in the manuscript. In these cases, we expect all author-generated code to be made available without restrictions upon publication of the work. Please review our guidelines at https://journals.plos.org/plosone/s/materials-and-software-sharing#loc-sharing-code and ensure that your code is shared in a way that follows best practice and facilitates reproducibility and reuse. 3. Thank you for stating in your Funding Statement: This research was supported by the German Research Foundation (SFB1027) and large-instrument funding under grant number INST 256/542-1 FUGG.  Please provide an amended statement that declares all the funding or sources of support (whether external or internal to your organization) received during this study, as detailed online in our guide for authors at http://journals.plos.org/plosone/s/submit-now.  Please also include the statement “There was no additional external funding received for this study.” in your updated Funding Statement. Please include your amended Funding Statement within your cover letter. We will change the online submission form on your behalf. 4. We note that Figures Fig 2G and Figure S3B-C in your submission contain copyrighted images. All PLOS content is published under the Creative Commons Attribution License (CC BY 4.0), which means that the manuscript, images, and Supporting Information files will be freely available online, and any third party is permitted to access, download, copy, distribute, and use these materials in any way, even commercially, with proper attribution. For more information, see our copyright guidelines: http://journals.plos.org/plosone/s/licenses-and-copyright. We require you to either present written permission from the copyright holder to publish these figures specifically under the CC BY 4.0 license, or remove the figures from your submission: a. You may seek permission from the original copyright holder of Figures Fig 2G and Figure S3B-C to publish the content specifically under the CC BY 4.0 license. We recommend that you contact the original copyright holder with the Content Permission Form (http://journals.plos.org/plosone/s/file?id=7c09/content-permission-form.pdf) and the following text:“I request permission for the open-access journal PLOS ONE to publish XXX under the Creative Commons Attribution License (CCAL) CC BY 4.0 (http://creativecommons.org/licenses/by/4.0/). Please be aware that this license allows unrestricted use and distribution, even commercially, by third parties. Please reply and provide explicit written permission to publish XXX under a CC BY license and complete the attached form.”Please upload the completed Content Permission Form or other proof of granted permissions as an "Other" file with your submission.  In the figure caption of the copyrighted figure, please include the following text: “Reprinted from [ref] under a CC BY license, with permission from [name of publisher], original copyright [original copyright year].” b. If you are unable to obtain permission from the original copyright holder to publish these figures under the CC BY 4.0 license or if the copyright holder’s requirements are incompatible with the CC BY 4.0 license, please either i) remove the figure or ii) supply a replacement figure that complies with the CC BY 4.0 license. Please check copyright information on all replacement figures and update the figure caption with source information. If applicable, please specify in the figure caption text when a figure is similar but not identical to the original image and is therefore for illustrative purposes only. 5. We notice that your supplementary figures are uploaded with the file type 'Figure'. Please amend the file type to 'Supporting Information'. Please ensure that each Supporting Information file has a legend listed in the manuscript after the references list.

Reviewers' comments:

Reviewer's Responses to Questions

**Comments to the Author**

1. Is the manuscript technically sound, and do the data support the conclusions?

Reviewer #1: Partly

Reviewer #2: Partly

2. Has the statistical analysis been performed appropriately and rigorously?

Reviewer #1: Yes

Reviewer #2: Yes

3. Have the authors made all data underlying the findings in their manuscript fully available?

Reviewer #1: Yes

Reviewer #2: Yes

4. Is the manuscript presented in an intelligible fashion and written in standard English?

Reviewer #1: Yes

Reviewer #2: Yes

**Reviewer #1:**  The authors have submitted a decently written manuscript describing the effects of a new compound miuraenamide A (MiuA), recently found to affect actin dynamics, on focal adhesions, cell contractility and migration. They show that in cell culture the treatment with MiuA in two cell types reduce average focal adhesion size but increase the number of adhesions. They also find that the strength of these adhesions, despite being higher in number, are less adhesive. The authors also show that amoeboid type migration is unaffected by MiuA treatment. Overall the experiments seem to be well-performed and nearly adequate analysis. However, the overall writing style needs some adjustment for clarity as well as some additional questions need to be addressed before this paper should be accepted for publication in PONE. The following issues should be addressed:

(1) The way the manuscript is written seems very redundant in many places. For example, the Introduction section is extremely long and is written to include the results. Provide only a quick summary of your results (not going figure by figure) in the introduction and keep the results section that you do have for the detailed explanation of the figures. On the first read of the manuscript, I thought that the introduction’s run through of the figures was indeed the results section.

(2) The authors use two cell types for the beginning two figure’s worth of experiments but then drop the use of the MEF cells. Why is this? MEF cells are commonly used for cell migration experiments, especially in 2D settings, so some analysis of their behavior upon MiuA treatment would be helpful to the field and further the impact of this paper.

(3) On a similar note, the authors don’t seem to address 2D migration parameters, which would be greatly beneficial to the broader cell migration community (and again, boost the impact of this publication). I suggest that the authors analyze the effect of MiuA treatment on 2D migration parameters such as speed, persistence, directionality, etc. These parameters are only analyzed in the context of the PDMS confinement channels during amoeboid migration. Is it known how MiuA affects unconfined 2D migration? Such a change in focal adhesions parameters surely affects 2D migration.

(4) In Figure 1, the authors show only the spreading area and actin filament length analysis for MEF cells but not RPE-1 cells – why? There seems to be a significant change in the spreading area for RPE-1 cells based on the images in panel (1C). Since the authors rely heavily on RPE-1 cells throughout the manuscript, this should not be omitted here.

**Reviewer #2:**  The authors demonstrate the effect of an actin-stabilizing agent (MiuA) on the adhesivity, contractility and migration of two cell types. They show that although focal adhesions are impacted by this drug, the contractility and ability to migrate in the amoeboid mode are not.

These findings are an interesting addition to a growing body of work on cell mechanics and cell-substrate mechanical interactions. However, some more mechanistic insights and explanation of the data could strengthen this study, and some of the rationale is not well-described. Some specific concerns and questions are:

1. The Introduction paragraphs 3 and 4 are essentially a summary of the Results, which is unconventional. As a reader, I am interested in knowing more about the rationale and background of the study. what is known about the biochemical mechanism of how MiuA works: does it inhibit actin polymerization or severing, for example? how does it interact with actin binding proteins? what are the pros and cons of using this as a stabilizng agent vs. jusplakinolide or phalloidin? what are the RPE1 and MEF cell types and why are they the suitable candidates in this study?

2. In Fig. 1, it is abrupt to start with quantification before showing cell images first. Also, why quantification is shown only for MEF cell type only in Figs 1a and 1b and not for RPE1?

3. Fig. 1c focuses on individual cells, but to be convincing the authors should also show a wider field of view with many cells, so that the reader can see the representative behavior.

4. The MiuA treated MEF cell shown has an interesting spindle-like morphology. Have the authros tried to quantify this effect? A quantification of the cell aspect ratio would reflect this, for example. It will be interesting to see how cell morphology is affected in addition to cell area, by MiuA.

5. There should be some explanation, at least qualitative, why the contractility is not affected although adhesivity is.

6. Why are neutrophils intriduced for motility measurements, and not MEF? do these not migrate?? there could be a cartoon to illustrate how persistence is measured. In addition to adhesions, can the 1D vs 2D nature of the migration also play a role?

7. Similar to Introduction, the Conclusion could be stronger if there was some discussion of the broader context of this work. How does jusplakinolide affect these quantities: cell adhesivity, contractility, and migration for example?

8. The images provided are low resolution at least in the reviewer's copy. Perhaps the images analyzed should also be shared in high resolution alongside the supplementary data.

9. There is some informal/colloquial use of English in some places that seem out of place in a technical document. Sentences should not begin with "And" as in line 63 or 88, and "very well" is not a quantifiable statement in line 76, just to take two examples. The authors could use better human proofreading in addition to using AI-based tools which are not often adequate for this purpose.

**Do you want your identity to be public for this peer review?** For information about this choice, including consent withdrawal, please see our Privacy Policy

Reviewer #1: No

Reviewer #2: No

---

## [Author Response · Author response to Decision Letter 1]

2 Jun 2025

We thank the editor and the reviewers for their time reading our manuscript and the patience while waiting for our response. We replied to all concernes and questions posed by the editor and the reviewers. Our replies can be seen in the attached files under "Reply to editor and reviwers.pdf".

---

## [Decision Letter · Decision Letter 1]

2 Jul 2025

Dear Dr. Lautenschläger,

Thank you for submitting your manuscript to PLOS ONE. After careful consideration, we feel that it has merit but does not fully meet PLOS ONE’s publication criteria as it currently stands. Therefore, we invite you to submit a revised version of the manuscript that addresses the points raised during the review process.

The manuscript was substantially improved by the revision and addressing the issues of both referees. However, before acceptance for publication it is imperative to address the last open points raised by referee 2.

We look forward to receiving your revised manuscript.

Kind regards,

Florian Rehfeldt

Academic Editor

PLOS ONE

Journal Requirements:

Reviewers' comments:

Reviewer's Responses to Questions

**Comments to the Author**

Reviewer #1: All comments have been addressed

Reviewer #2: (No Response)

2. Is the manuscript technically sound, and do the data support the conclusions?

Reviewer #1: Yes

Reviewer #2: Yes

3. Has the statistical analysis been performed appropriately and rigorously?

Reviewer #1: Yes

Reviewer #2: Yes

4. Have the authors made all data underlying the findings in their manuscript fully available?

Reviewer #1: Yes

Reviewer #2: Yes

5. Is the manuscript presented in an intelligible fashion and written in standard English?

Reviewer #1: Yes

Reviewer #2: Yes

Reviewer #1: The authors have satisfied all of my concerns. I recommend an acceptance of this manuscript for publication.

Reviewer #2: The authors have significantly improved the clarity of the manuscript in this revised version. They have now answered most of my questions, but I have some more remaining comments which I believe should be addressed before final acceptance.

1.The authors explain that the some of the quantification for RPE-1 cells, corresponding to the results in Fig. 1, were carried out in their previous publication, Ref.22. However, in that study, the MiuA-treated RPE-1 cells increased their spread area, whereas MiuA-treated MEF cells showed no change in spread area. They think this puzzling result is because different doses of MiuA were used for the two cell lines. This raises the obvious question: why not do the same dosage for both cell lines to establish this? Is there any practical limitation to this?

2. Related to above, the authors discuss that one of the disadvantages of using jasplakinolide is that its effect is highly dosage dependent. However, they also found that MiuA's effect is also dosage dependent. How sensitive is this dosage dependence? Have they tried for example, different doses of MiuA on RPE-1 or MEF?

3. In Fig. 1a, the MiuA treated MEF cells shown look distinctly smaller by eye than the control MEF cells. Why is this? Is this image not a representative sample of the whole population of cells? In this case, it might be more convincing to choose a different image, or show additional images in the SI.

**Do you want your identity to be public for this peer review?** For information about this choice, including consent withdrawal, please see our Privacy Policy

Reviewer #1: No

Reviewer #2: No

---

## [Author Response · Author response to Decision Letter 2]

23 Jul 2025

We thank the editor and the reviewers for their time reading our manuscript and the patience while waiting for our response. We replied to all concernes and questions posed by the editor and the reviewers. Our replies can be seen in the attached files under "Reply to editor and reviewers.pdf".

---

## [Decision Letter · Decision Letter 2]

27 Jul 2025

Modulation of Cellular Adhesion, Contractility, and Migration by MiuA: A Comprehensive Analysis of Its Biomechanical Impact

PONE-D-24-57233R2

Dear Dr. Lautenschläger,

We’re pleased to inform you that your manuscript has been judged scientifically suitable for publication and will be formally accepted for publication once it meets all outstanding technical requirements.

Kind regards,

Florian Rehfeldt

Academic Editor

PLOS ONE

Additional Editor Comments (optional):

Reviewers' comments:

Reviewer's Responses to Questions

**Comments to the Author**

Reviewer #2: All comments have been addressed

2. Is the manuscript technically sound, and do the data support the conclusions?

Reviewer #2: Yes

3. Has the statistical analysis been performed appropriately and rigorously?

Reviewer #2: Yes

4. Have the authors made all data underlying the findings in their manuscript fully available?

Reviewer #2: Yes

5. Is the manuscript presented in an intelligible fashion and written in standard English?

Reviewer #2: Yes

Reviewer #2: The authors have shown additional data and satisfactorily answered my remaining questions about dosage dependence of miuA and additional cell images. I think this is technically sound and interesting work, and is ready to be accepted for publication.

**Do you want your identity to be public for this peer review?** For information about this choice, including consent withdrawal, please see our Privacy Policy

Reviewer #2: No

---

## [Editor Report · Acceptance letter]

PONE-D-24-57233R2

PLOS ONE

Dear Dr. Lautenschläger,

I'm pleased to inform you that your manuscript has been deemed suitable for publication in PLOS ONE. Congratulations! Your manuscript is now being handed over to our production team.

Kind regards,

on behalf of

Dr. Florian Rehfeldt

Academic Editor

PLOS ONE